# The effects of aging and musicianship on the use of auditory streaming cues

**Sarah A. Sauvé** [1]*, **Jeremy Marozeau** [2], **Benjamin Rich Zendel**[1]

**1** Division of Community Health and Humanities, Faculty of Medicine, Memorial University of Newfoundland, St. John's, Newfoundland and Labrador, Canada, **2** Department of Health Technology, Technical University of Denmark, Lyngby, Denmark

* sarah.sauve@mun.ca

**Data Availability Statement:** Aggregated data is available in Supplementary Materials 2.

**Funding:** This research was supported by BZR's Canada Research Chair. https://www.chairs-chaires.gc.ca/home-accueil-eng.aspx The funders

## Abstract

Auditory stream segregation, or separating sounds into their respective sources and tracking them over time, is a fundamental auditory ability. Previous research has separately explored the impacts of aging and musicianship on the ability to separate and follow auditory streams. The current study evaluated the simultaneous effects of age and musicianship on auditory streaming induced by three physical features: intensity, spectral envelope and temporal envelope. In the first study, older and younger musicians and non-musicians with normal hearing identified deviants in a four-note melody interleaved with distractors that were more or less similar to the melody in terms of intensity, spectral envelope and temporal envelope. In the second study, older and younger musicians and non-musicians participated in a dissimilarity rating paradigm with pairs of melodies that differed along the same three features. Results suggested that auditory streaming skills are maintained in older adults but that older adults rely on intensity more than younger adults while musicianship is associated with increased sensitivity to spectral and temporal envelope, acoustic features that are typically less effective for stream segregation, particularly in older adults.

## 1 Introduction

In everyday life, our brain continuously analyses and organizes the omnipresent mixture of sounds reaching our eardrums. This organizational process is called auditory scene analysis (ASA; 1). In his seminal book, Bregman [1] describes ASA as the process by which we integrate and segregate acoustic information to form perceptual objects/streams that accurately represent the acoustic environment. ASA combines physical features from the environment (i.e., bottom-up), and learned knowledge (i.e., top-down) to segregate and track sound sources over time. Many bottom-up sound cues affect how efficiently this analysis can be carried out. As a general rule, when physical features suggest the presence of multiple sound sources (e.g., differing onsets, spectral components whose frequencies are not integer multiples of a fundamental frequency (F0), differing inter-aural time/levels, etc...), it is more likely for the sounds to be perceptually segregated into separate perceptual streams. When physical features suggest that sounds come from the same source (e.g., similar onsets, spectral components whose

had no role in study design, data collection and analysis, decision to publish or preparation of the manuscript.

**Competing interests:** The authors have declared that no competing interests exist.

frequencies are integer multiples of a F0, similar inter-aural time/levels, etc. . .) it is more likely that they will be perceptually integrated into a single perceptual stream. A combination of bottom-up sound features, such as frequency [1, 2], tempo [2, 3], location [4, 5], and temporal and spectral envelopes [6–9], can contribute to the quality and efficiency of auditory stream segregation. Top-down cognitive aspects such as attention [10], expectation [11] and musicianship [12, 13] also influence the accuracy and efficiency of stream segregation.

The separation of two sound sources into streams involves both simultaneous and sequential aspects. Evidence suggests that aging negatively affects concurrent stream segregation, or the ability to segregate simultaneous sounds [14–18]. On the other hand, sequential stream segregation–the segregation of sequential sounds–is little affected by age [19–22], even for participants with hearing loss [23, 24]. This pattern suggests that older adults have more difficulty than younger adults separating sounds that occur simultaneously. Still, once the sounds are separated, older adults can track a sequence of sounds as well as younger adults.

Auditory stream segregation is important when listening to music. For example, there are sections of J.S. Bach's Toccata and Fugue in D minor where sequential notes that differ in frequency are played rapidly in order to give the perception that there are two simultaneous melodies. In this situation, the participant hears a rapid change in F0 between the notes that suggests there must be two sound sources producing the music. In this case, the resulting percept is that of two auditory streams or two melodies. Rapidly alternating tones that differ in frequency are often used to explore auditory stream segregation in the lab. For example, Snyder and Alain [20] investigated potential age-related effects on sequential auditory streaming using the ABA_ paradigm, where 'A' and 'B' represent tones that differ in frequency and '_' represents silence. When the A and B tones differ by a small amount in frequency, a 'galloping' rhythm is heard. As the frequency separation between the A and B tones increases, the likelihood of hearing two simultaneous streams increases. Snyder and Alain [20] presented young, middle-aged and older participants with ABA_ sequences where the A and B tones were 0, 4, 7 or 12 semitones apart (the A tone frequency was 500 Hz and the B tone frequencies were 500, 625, 750 or 1000 Hz) and asked them to indicate whether they perceived one stream (galloping percept) or two streams (isochronous percept). All participants had near-normal pure-tone average (PTA) thresholds at low frequencies (PTA0.5-1kHz < 30 dB HL), but at higher frequencies, the older adults had significantly higher thresholds than the younger adults (e.g., PTA at 8000Hz in the left ear was 5 dB HL [SD = 7.5 dB] for the younger adults, and was 66 dB HL [SD = 20.5 dB] for the older adults). These PTA differences are generally considered a normal part of aging [20, though see 25]. Despite these PTA differences between older and younger adults, there was no difference in streaming perception between the groups. This may be because the stimuli were all below 1000 Hz, and both groups of participants had normal thresholds at those frequencies. All participants exhibited a similar tendency to perceive two simultaneous streams as the frequency separation between the A and B tones increased. Snyder and Alain [20] only assessed the effect of frequency separation on auditory streaming, but there are many other acoustic features that can give rise to stream segregation, including harmonic structure (timbre), and intensity [26].

The effects of aging on other acoustic features that give rise to sequential auditory streaming are not as well known. The processing of temporal aspects of acoustic stimuli is reduced in older adults [27, 28]. Behavioural evidence suggests that older adults have more difficulty than younger adults identifying the temporal order of sound sequences [29–32] and performing temporal discrimination tasks, such as silent gap detection [33–36]. Interestingly, hearing loss had little impact on these abilities, as performance on these tasks was similar for older adults with and without hearing loss [29, 30, 33–36]. Evidence from neuroscience suggests that the

encoding precision of temporal information is poorer in older adults with normal hearing than in younger adults [37, 38].

Little is known about how aging affects the encoding of a sound's spectral envelope. One study [39] tested melody and timbre recognition for a variety of stimulus manipulations, including low- and high-pass filters, vocoding and temporal envelope modulation. While younger adults were able to use both low- and high-pass modified stimuli to identify timbre, older adults with normal to near-normal (PTA0.25-4kHz < 25 dB HL) hearing were not able to use the higher frequency information. In another study, Grimault et al. [40], used an ABA task with complex tones where the F0 of the A tone was either 88 Hz or 250 Hz and the F0 of the B tone varied between 88 Hz and 352 Hz and asked younger and older adults with moderate or mild hearing loss to indicate whether they perceived one or two streams at the end of a 4s segment. They found that older adults with moderate hearing loss (PTA0.5-2kHz between 32 and 72 dB HL) and with mild hearing loss (PTA0.5-2kHz between 30 and 35 dB HL) had lower d' scores than younger adults when the A tone's F0 was 250 Hz, but not 88 Hz, suggesting difficulty differentiating tones with higher frequencies, and therefore higher harmonics. It is possible that these higher harmonics were not resolved due to broader auditory filters [41], which are specifically associated with hearing loss rather than increased age [42]. Studies specifically investigating the effect of hearing loss on stream segregation showed that streaming is impaired in older adults with hearing loss [43]. More specifically, auditory streaming induced by frequency [18, 23, 40, 44] and inter-aural time differences [45] is worse for older adults with hearing loss than for older adults with normal hearing. Overall, it is likely that aging and hearing loss affect the ability to discriminate the acoustic features needed to form auditory streams but do not affect the ability to stream sounds that are perceptually distinct.

While aging can negatively affect auditory stream segregation, musicianship seems to have a positive effect. One of the first studies to explore this musician advantage for stream segregation investigated the time decay of auditory stream biasing by playing a 10 second sequence of repeated A tones, called an induction sequence, a silent interval of between 0 and 8 seconds and a short ABAB test sequence, where participants indicated whether they heard one or two streams. Beauvois and Meddis [12] found that auditory streams persisted over longer silences for musicians than for non-musicians. More recently, Marozeau et al. [7] investigated the effect of musicianship on stream segregation. Musicians (mean age 31 years, SD = 7.2 years) and non-musicians (mean age 32.2 years, SD = 7.9 years) listened to a sequence of tones in which a four-note repeating *target pattern* was 'hidden' amongst *distractor tones* (see Fig 1). The distractor tones were manipulated in terms of intensity, spectral envelope and temporal envelope so that they varied in similarity. Participants were asked to continuously rate how easily they could perceive the target pattern. In a dissimilarity paradigm, participants performed perceptual similarity ratings for pairs of stimuli varying in intensity, spectral envelope and temporal envelope. These ratings were used to map the stimuli into one multi-dimensional space, allowing the effects of the three manipulated features to be directly compared to each other on a common perceptual scale. Compared to non-musicians, the musicians needed less physical difference between the targets and the distractors to successfully segregate the two streams. In terms of acoustic cues, intensity, led to the most robust stream segregation for musicians while intensity and spectral envelope were similar in effectiveness for non-musicians. This study suggests that a participant's experience and knowledge (i.e., top-down) can affect how acoustic cues (i.e., bottom-up) are utilized during auditory stream segregation.

The goal of the present study was to explore how the interaction of musicianship, aging, and audiometric thresholds affect the relative salience of intensity, spectral envelope and temporal envelope as auditory streaming cues, using a modified version of the target/distractor paradigm described above [46, 47].

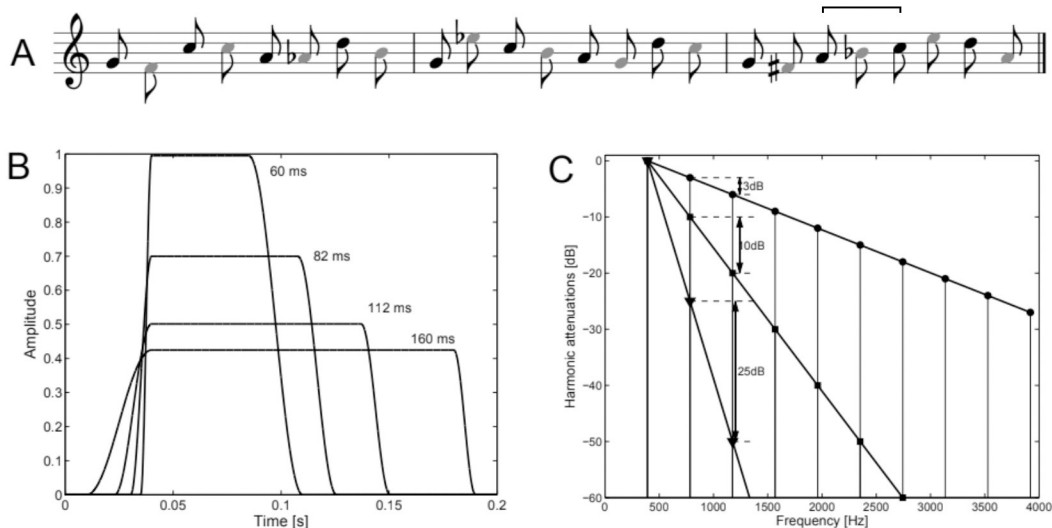

**Fig 1. Illustration of stimuli and feature manipulations.** (A) Sample stimuli, where black note heads depict the target pattern while grey note heads depict the distractor sequence. An example of a deviant target is given in the third measure, where the two notes marked by the bracket were reversed in order. (B) Temporal envelopes varying in impulsiveness from 60 ms FDHM (full duration at half-maximum) to 160 ms FDHM (the target notes). Only 4 of the 20 possible temporal envelopes are shown here. (C) Spectral envelopes varying from 3 dB of attenuation per harmonic (the target notes—circles) to 25 dB of attenuation per harmonic (triangles). Only 3 out of the 20 possible spectral envelopes are shown here. Fig adapted from Marozeau et al. (2013), with permission.

Based on current evidence of the maintenance of sequential streaming abilities in older adults [19, 20] and the positive relationship between musicianship and (non-speech) stream segregation in younger adults [12, 48–51], we expected older musicians to require a similar level of dissimilarity between the target and the distractor to successfully segregate the two streams as young musicians [52–56], and less dissimilarity than older and younger non-musicians. Of particular interest here is the interaction between age and musicianship. It is unknown whether the positive relationship between musicianship and auditory streaming is consistent throughout the lifespan. This auditory streaming task can also be interpreted as an inhibition task, where the distractor tones must be successfully separated and ignored in order to perform the task. Though the inhibition theory of aging suggests a decline in inhibition [57–59], according to a recent meta-analysis of inhibition in aging, the ability to ignore distracting information remains intact in older adults [60], supporting the hypothesis that older and younger adults will perform similarly on the current streaming task; however, given that temporal encoding and discrimination decrease with increasing age [32, 34], streaming based on temporal envelope in the current study should be more difficult for older adults than younger adults. Similarly, as age-related hearing loss is usually high-frequency hearing loss and the manipulation of spectral envelope in this study involved varying attenuation of the higher frequencies of the distractor tones, loss of hearing in this range may lead to poorer timbral differentiation and therefore poorer streaming and task performance when spectral envelope is manipulated. Accordingly, both age and hearing loss will be important variables to consider independently.

To test these hypotheses, two experiments were conducted. In the first experiment, younger and older musicians and non-musicians identified deviations in the target pattern while distractor similarity to the target either increased or decreased over time. This modification provided a more objective measure of auditory streaming than a subjective perceptual rating (or

judgment, 20), as identifying a deviant is only possible if the target is streamed from the distractor tones. In a second experiment, as in Marozeau et al. [7], younger and older musicians and non-musicians rated the similarity of pairs of melodies differing in intensity, spectral envelope and temporal envelope. Using multidimensional scaling (MDS), a common perceptual dissimilarity scale between the three manipulated features was extracted to allow a direct comparison of the effects of each of the acoustic features.

## 2 Experiment 1

### 2.1 Method

**2.1.1 Participants.** Fifty-four participants were recruited and provided written informed consent following the Interdisciplinary Committee on Ethics in Human Research at Memorial University of Newfoundland (ICEHR#20192257). Participants were divided into two age groups: 28 *younger adults* (<38 years; age range 17–37; 16 female) and 26 *older adults* (> 60 years; age range 61–82; 9 female). Musicianship was measured using the Goldsmiths Musical Sophistication Index's musical training sub-scale (58; henceforth referred to as the Gold-MSI, see section 2.1.2). Participants varied widely in their musical training backgrounds. All participants self-reported being healthy and free of any cognitive deficit. Hearing abilities were assessed using pure-tone audiometry. The reported pure-tone average (PTA) is based on thresholds at 500, 1000, 2000 and 4000 Hz for the better-hearing ear. Recently updated WHO guidelines define normal hearing as a PTA (i.e., average across 0.5, 1, 2 and 4 kHz) below 20 dB HL, while mild hearing loss is defined as a PTA between 20 and 34 dB HL [61]. Previous WHO guidelines considered normal hearing to be below 26 dB HL [61] and participants in the current study were screened with this criterion in mind. Thus, our sample only includes people with very mild hearing loss (i.e., less than 26 dB HL PTA). In a sample of older adults, it is normal to find many participants with mild hearing loss, with multiple large epidemiological studies showing that around 70% of older adults experience at least mild hearing loss by their late 70's [62]. All participants received a $20 honorarium for their participation. Table 1 summarizes participant demographics.

**2.1.2 Gold-MSI.** The Goldsmiths Musical Sophistication Index (Gold-MSI) [63] musical training sub-scale consisted of 7 questions, each answered on a 7-point Likert scale, for a total score out of 49. Two questions were subjective (e.g. 'I would not consider myself a musician') and five asked about length in years of formal instrumental and music theoretical training and

**Table 1. Participant demographics–Experiment 1.**

| | Age[a] | Gold-MSI training sub-scale score [b] | Pure-tone Average (PTAv; dB HL)[c] |
|---|---|---|---|
| Younger Adults; Musicians | 25.9 (5.4) | 38.9 (7.0) | 1.5 (2.5) |
| Younger Adults; Non-musicians | 26.4 (5.5) | 12.9 (5.5) | 4.9 (6.7) |
| Older Adults; Musicians | 70.0 (5.1) | 35.9 (4.4) | 14.0 (9.9) |
| Older Adults; Non-musicians | 68.4 (6.3) | 14.8 (6.7) | 15.7 (7.8) |

[a]$t(51)$ = -28.85, $p < .01$ between age groups; standard deviation in brackets

[b]$t(51)$ = 14.20, $p < .01$ between musical training groups; standard deviation in brackets

[c]Better ear average of pure-tone threshold at 500, 1000, 2000 & 4000 Hz; $t(39)$ = 5.78, $p < .01$ between age groups; standard deviation in brackets

length in years and hours per day of regular practice (e.g. At the peak of my interest, I practiced __ hours a day on my primary instrument). The sub-scale additionally asked the participant to report which instrument they played best. This continuous factor derived from an individual's music training history will be referred to as *musical training*. For visualization and summary purposes, musicians were defined as participants scoring above 50% on the Gold-MSI (i.e. a score > 25; 14 younger, 12 older) and non-musicians as scoring less than 50% (i.e. a score < 25; 14 younger, 13 older). Note that no participant scored exactly 25. Though it is possible that differences in task performance between participants with varying musical training backgrounds may be attributed to differences other than formal musical training (e.g. genetics), in this study musical training history is the measured proxy for musicianship.

**2.1.3 Stimuli.** Stimuli were identical to those of Marozeau et al. [7], Experiment 1. Fig 1 illustrates the stimuli, which consisted of two patterns: one four-note repeating melody (*target*; black note heads) with notes G4, C5, A5 and D5, and pseudorandom notes (*distractor*; grey note heads). The tones of the target melody had a temporal envelope consisting of a 30 ms raised-cosine onset, 140 ms of sustain, and a 10 ms offset, and a spectral envelope with 10 harmonics, successively attenuated by 3 dB. The F0s of the distractor notes were uniformly randomly selected from a range of an octave around the target melody (F4 to E5) and were manipulated along one feature at a time while the remaining features were kept constant. Loudness level, spectral envelope and temporal envelope were varied over 20 degrees of difficulty: at difficulty 1, they allowed easy segregation and at difficulty 20 they were the same as the target melody.

The stimuli of Marozeau et al. [7] were designed to be presented in free field at a loudness level of 65 phons (as loud as 1 kHz tone at 65 dB SPL), using a loudness model [64], to ensure that the sensation of loudness was constant even though the spectral and temporal envelope were varied. Stimuli are described with the specifications with which they were designed.

*2.1.3.1 Intensity.* Intensity was attenuated, in twenty steps of 2 phons, from equal level to target to an attenuation of 38 phons. The starting difficulty (no attenuation or 38 phons) of the distractor depended on the trial block (details in section 2.1.4 below).

*2.1.3.2 Temporal envelope.* Tone impulsiveness, defined as the full duration of the sound at half of the maximum amplitude (FDHM), was logarithmically spaced over twenty degrees of difficulty from 160 ms (equal to target) to 60 ms (Fig 1B), where the starting difficulty depended on the trial block.

*2.1.3.3 Spectral envelope.* The spectral envelope attenuation was manipulated in twenty logarithmically spaced steps from 3 dB (equal to target) to 25 dB attenuation per harmonic (Fig 1C). Note that at 25 dB attenuation per harmonic, the majority of harmonics were inaudible. The starting difficulty of the distractors' spectral envelope depended on the trial block.

The stimuli were constructed using Matlab 7.5 and sequences were generated using MAX/MSP 8. For further details, see Marozeau et al. [7], Experiment 1, Stimuli subsection. For practical reasons, in this experiment sounds were played through over-ear headphones (Sennheiser HDA 200) at a comfortable level set by the listeners and ranging from 25–60 dB SPL. This affects the absolute intensity at which the stimuli were presented, which may affect baseline performance; however, the manipulation of interest is the difference in intensity between the target and the distractors at which the deviant detection task (described below) becomes possible, not the absolute intensity.

**2.1.4 Procedure.** The experiment consisted of six blocks, three based on increasing difficulty and three based on decreasing difficulty for each of the acoustic features. In each of these blocks, the target melody was presented in a continuous loop while distractor tones were interleaved. Two of the target melody tones were reversed in order 25% of the time, creating a *deviant* melody. The participants' task was to press the space bar when they detected such a

deviant. Good performance is only possible if the target and distractor tones are perceived as separate streams. The deviant tones in each block varied along a single feature and each feature was tested twice, once with *increasing* difficulty (least similarity between target melody and distractor to exact match between target melody and distractor) and once with *decreasing* difficulty (exact match between target melody and distractor to least similarity between target melody and distractor). Blocks varied in difficulty according to the following adaptive rule: three consecutive *hits*, or correct identifications of a deviant, or three consecutive *misses* (i.e. failing to identify the deviant) were necessary to move onto a new degree of difficulty. When difficulty was increasing, three consecutive hits or misses were needed to move to a higher degree of difficulty (more similarity between target melody and distractor), while a complete lack of hits for three successive difficult levels led to the manual termination of the block. When difficulty was decreasing, three consecutive hits or misses were required to move to a lower degree of difficulty (less similarity between target melody and distractor). In these decreasing blocks, participants heard all stimulus degrees of difficulty. Movement between degrees of difficulty only occurred in one direction, based on whether difficulty was increasing or decreasing.

A response within 1.5 seconds (six tones, target or distractor) of a deviant was considered a hit. Any other response was considered a false alarm. There was no limit on the number of *false alarms*, or the identification of a deviant where there was none, leading to a variable amount of time spent at each degree of difficulty. A practice block was always completed first, and included the first few degrees of difficulty for the increasing intensity manipulation, until the participant understood the task; data from this practice block were discarded. Based on piloting, in an effort to minimize false alarms, participants were instructed to only identify a deviant if they could clearly perceive the target melody. They were reassured that it was normal not to perceive the target melody at all in the early stages of a block manipulated with decreasing difficulty and to simply wait until the target melody could be distinguished from the distractors. Despite this, false alarms occurred often, as discussed in Section 2.3.1.

**2.1.5 Analysis.** All data analysis was performed in R 3.3.2. Alpha was set at .01, with the conservative Bonferroni correction applied for multiple comparisons. Effect sizes are reported for all statistical tests. Transformed data and analysis scripts are available from the first author upon request. Summary statistics can be found in the S2 Appendix.

*2.1.5.1 Data processing.* Raw data were transformed into *d'* scores for each degree of difficulty, feature and participant. One older non-musician participant's data were removed as they were unable to perform the task after the first few degrees of difficulty on more than half the blocks. Effects of block order and intensity at which the stimuli were played were tested as potential confounding variables using mixed effects multiple linear regression models. Both were non-significant and consequently excluded from the omnibus test described below.

*2.1.5.2 Omnibus test.* Mixed effects multiple linear regression models as implemented by the *lme4* package [65] were used to analyse *d'* values for each *feature*. In this model, fixed effects included *degree of difficulty*, *age group*, *musical training* and *pure tone average (PTA)* with random intercepts on participants. *Degree of difficulty* and *age group* were categorical variables, coded as factors, where *Difficulty 1* and *younger* were the factor level against which all other factor levels were compared. *Musical training* and *PTA* were continuous variables. The model was evaluated using Pearson's correlation between the model's predictions and the data along with the correlation's 95% confidence interval and effect size $R^2$. The statistical significance of each predictor was determined using the *lmer* function's default t-tests and p-values, which employ the Satterthwaite method [66]. All follow-up pairwise comparisons used between-samples t-tests with Bonferroni correction.

*2.1.5.3 Equivalence test.* A two-one-sided t-test (TOST) procedure was applied using the *TOSTER* package [67] in cases where the age effect was null. An equivalence test detects whether an effect is statistically different from zero and whether an effect is larger than a set smallest interesting effect size, or equivalence bound. In other words, in the presence of a null effect according to the omnibus test, the TOST procedure assesses whether the effect is non-zero and if it is large enough to be considered interesting. Here, the equivalence bound, or smallest interest effect size was set to Cohen's *d* of 0.2.

## 2.2 Results

Task performance as measured by *d'* is illustrated in Fig 2. The mixed effects multiple linear regression models are summarized in Tables 2 (intensity), 3 (spectral envelope) and 4 (temporal envelope), where only significant main effect predictors are included for brevity. Full model specifications can be found in the S1 Appendix.

**2.2.1 Intensity.** The model for intensity included significant main effects of *degree of difficulty* and *musical training*, with no significant interactions. Pairwise comparisons between degrees of difficulty indicated that *d'* was significantly different (Bonferroni correction $p < 5 \times 10^{-5}$) between pairs of lower and higher degrees of difficulty, as illustrated in Fig 2A (details of all tests can be found in the S1 Appendix). Participants with higher musical training scores were better at detecting deviants overall. To confirm the null effect of age, a TOST comparing younger and older adults with lower and upper equivalence bounds of *d'* = -0.40 and *d'* = 0.40, equivalent to a Cohen's *d* of 0.2. The equivalence test was not significant, $t(2098) = 1.62$, p > .01, indicating that the effect, $\Delta d' = -0.26$ CIs [-0.46, -0.06], was in fact different from zero, where older adults had higher *d'* scores than younger adults but this difference did not reach significance in the omnibus test.

**2.2.2 Spectral envelope.** The model for spectral envelope included a significant main effect of *degree of difficulty* and *musical training* with no significant interactions. Pairwise

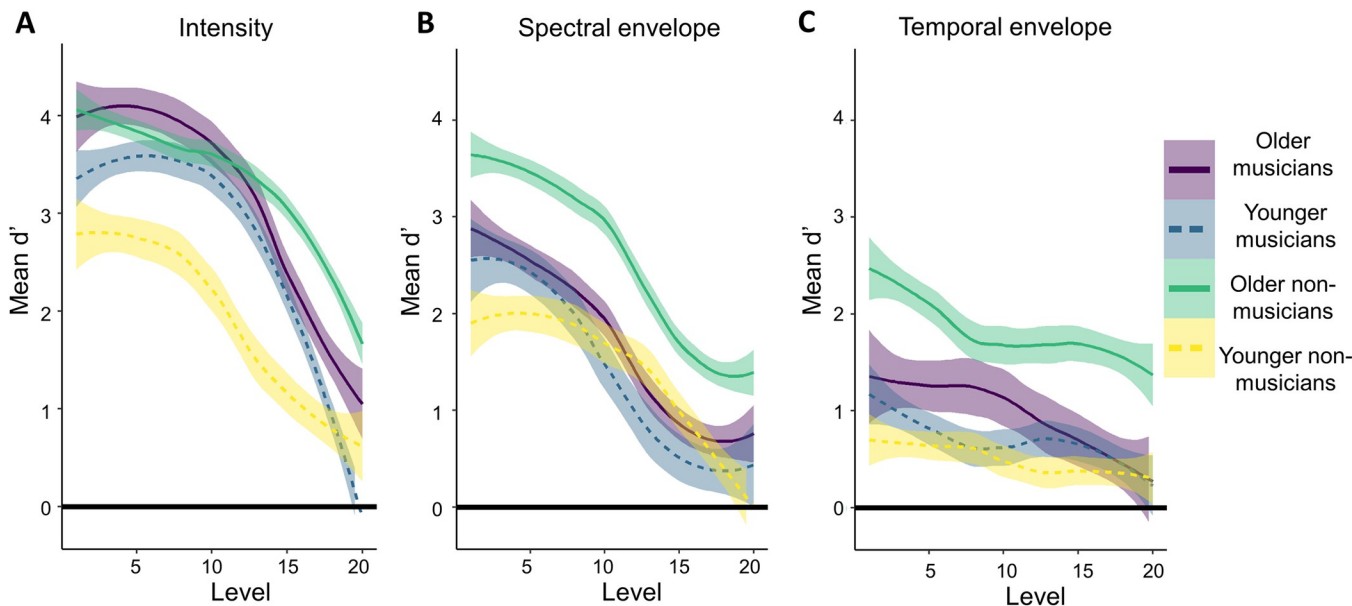

**Fig 2. Mean *d'* scores.** Mean *d'* score by degree of difficulty, grouped by age group and musical training for intensity (A), spectral envelope (B) and temporal envelope (C) features. Standard error is shown by grey shading around each line. While musical training was a continuous variable in statistical analysis, for visualization purposes, participants with scores over 25 were considered musicians, and participants with scores less than 25 were considered non-musicians.

**Table 2. Mixed effects multiple linear regression model for intensity feature.** Coefficients for each predictor (and each degree of difficulty, as relevant) along with standard error (SE) and predictor $R^2$ are reported along with Pearson's correlation, 95% CIs, p-value and $R^2$ for full models in the note below.

| Intensity | | | |
|---|---|---|---|
| **Predictor** | **Coefficient** | **SE** | **$R^2$** |
| Intercept | 2.36 | 0.60 | .31 |
| Difficulty 2 | -0.01 | 0.56 | .22 |
| Difficulty 3 | -0.11 | | |
| Difficulty 4 | -0.13 | | |
| Difficulty 5 | -0.31 | | |
| Difficulty 6 | -0.65 | | |
| Difficulty 7 | -0.37 | | |
| Difficulty 8 | -0.73 | | |
| Difficulty 9 | -0.29 | | |
| Difficulty 10 | -0.27 | | |
| Difficulty 11 | -1.96 | | |
| Difficulty 12 | -1.88 | | |
| Difficulty 13 | -2.11 | | |
| Difficulty 14 | -2.35 | | |
| Difficulty 15 | -2.16 | | |
| Difficulty 16 | -2.48 | | |
| Difficulty 17 | -2.44 | | |
| Difficulty 18 | -2.52 | | |
| Difficulty 19 | -2.45 | | |
| Difficulty 20 | -2.38 | | |
| Musical training | 0.03 | 0.02 | < .01 |

Note. r = .75, CIs [.73, .76], p < .01, $R^2$ = .56

comparisons between degrees of difficulty indicated that $d'$ was significantly different (Bonferroni correction $p < 5 \times 10^{-5}$) between pairs of lower and higher levels, as illustrated in Fig 2B (details of all tests can be found in the S1 Appendix). Participants with higher musical training scores were better at detecting deviants overall, including when the difference in spectral envelope between target and distractor was smaller. To confirm the null effect of age, a TOST comparing younger and older adults with lower and upper equivalence bounds of $d'$ = -0.42 and $d'$ = 0.42. The equivalence test was not significant, $t$ (2109) = 0.21, p > .01, indicating that the effect, $\Delta d'$ = 0.44, CIs [0.23, 0.65], was non-zero and fell outside our equivalence bounds. Overall, an effect of age was not significant according to the omnibus test but was larger than the smallest interesting effect.

To investigate whether this ambiguity was due to an undetected interaction between age and musical training, two further TOST were applied. First, a TOST compared younger non-musicians and all older adults with lower and upper equivalence bounds of $d'$ = -0.40 and $d'$ = 0.40. The equivalence test was not significant, $t$ (1185) = 2.26, p > .01. The effect, $\Delta d'$ = -0.16, CIs [-0.41, 0.08], fell within our equivalence bounds but its CI extended beyond it; however, the CI for this effect also included zero, indicating that the difference between younger non-musicians and older participants was inconclusive. Second, a TOST compared younger musicians and all older adults with lower and upper equivalence bounds of $d'$ = -0.42 and $d'$ = 0.42. The equivalence test was not significant, $t$ (1113) = 5.09, p > .01, and our effect, $\Delta d'$ = 0.98, CIs [0.72, 1.24] fell outside our equivalence bounds. While the TOST comparing younger non-

**Table 3. Mixed effects multiple linear regression model for spectral envelope feature.** Coefficients for each predictor (and each degree of difficulty, as relevant) along with standard error (SE) and predictor $R^2$ are reported along with Pearson's correlation, 95% CIs, p-value and $R^2$ for full models in the note below.

| Spectral envelope | | | |
|---|---|---|---|
| **Predictor** | **Coefficient** | **SE** | **$R^2$** |
| Intercept | 1.20 | 0.67 | .35 |
| Difficulty 2 | 0.46 | 0.62 | .16 |
| Difficulty 3 | -0.04 | | |
| Difficulty 4 | 0.03 | | |
| Difficulty 5 | 0.26 | | |
| Difficulty 6 | 0.56 | | |
| Difficulty 7 | -0.22 | | |
| Difficulty 8 | -0.15 | | |
| Difficulty 9 | -0.10 | | |
| Difficulty 10 | -0.61 | | |
| Difficulty 11 | 0.04 | | |
| Difficulty 12 | -0.37 | | |
| Difficulty 13 | -0.38 | | |
| Difficulty 14 | -0.63 | | |
| Difficulty 15 | -1.03 | | |
| Difficulty 16 | -1.29 | | |
| Difficulty 17 | -2.09 | | |
| Difficulty 18 | -1.79 | | |
| Difficulty 19 | -1.60 | | |
| Difficulty 20 | -2.42 | | |
| Musical training | 0.05 | 0.02 | < .01 |

Note. r = .73, CIs [.71, .75], p < .01, $R^2$ = .53

musicians to older adults was inconclusive, the TOST comparing younger musicians to older adults showed that this difference was non-zero.

**2.2.3 Temporal envelope.** The model for temporal envelope included a significant main effect of *musical training*, with no significant interactions. Results are illustrated in Fig 2C. Details of all tests can be found in the S1 Appendix. To confirm the null effect of age, a TOST comparing younger and older adults with lower and upper equivalence bounds of *d'* = -0.34 and *d'* = 0.34, equivalent to a Cohen's *d* of 0.2. The equivalence test was not significant, *t* (2100) = 0.66, p > .01, indicating that the effect, *Δd'* = 0.39 CIs [0.22, 0.56], was different from zero, where younger adults had higher *d'* scores than older adults but this difference did not reach significance in the omnibus test.

**Table 4. Mixed effects multiple linear regression model for temporal envelope feature.** Coefficients for each predictor (and each degree of difficulty, as relevant) along with standard error (SE) and predictor $R^2$ are reported along with Pearson's correlation, 95% CIs, p-value and $R^2$ for full models in the note below.

| Predictor | Coefficient (SD) | SE | $R^2$ |
|---|---|---|---|
| Intercept | -0.04 | 0.57 | .44 |
| Musical training | 0.06 | 0.02 | < .01 |

Note. r = .69, CIs [.67, .72], p < .01, $R^2$ = .48

## 2.3 Discussion

**2.3.1 Results summary.** This study aimed to investigate auditory streaming abilities in terms of age and its interaction with musicianship for three sound features. Younger and older musicians and non-musicians identified deviants randomly inserted in a repeated four-note melody and interleaved with distractor tones that were more or less similar to the melody tones in intensity, spectral envelope and temporal envelope. The ability to detect the deviants was affected by the manipulation of the intensity, spectral envelope and temporal envelope of the distractor tones. This supports the idea that the acoustic manipulations affected auditory stream segregation and replicates previous findings [7]. More importantly, people with higher musical training scores were better able to make use of all features of the distractor at all levels of difference from the target.

Critically, though some literature suggests an effect of hearing loss as little as 10–15 dB on auditory discrimination [68, 69], PTA was never a significant predictor, suggesting that stream segregation based on intensity, spectral or temporal envelope was not affected by the variation in hearing thresholds in this study. This is consistent with literature supporting the preservation of sequential auditory streaming for both normal hearing and hearing-impaired older adults based on frequency and interaural time differences [19–21, 45]. Performance was also not significantly related to differences in the intensity at which stimuli were presented through the headphones, suggesting that there are not significant differences in baseline performance as a function of absolute intensity.

**2.3.2 Effect of musicianship.** The positive effect of musical training on auditory streaming is consistent with existing literature [48–51] and replicates Marozeau et al.'s [7] findings. Our model coefficients show that the Gold-MSI musical training subtest score was positively correlated with *d'*, corresponding to a *d'* advantage of roughly 0.6 between our musician and non-musician groups when intensity was manipulated and 1 when spectral envelope was manipulated. When temporal envelope was manipulated, there was a significant main effect of musical training only, corresponding to a *d'* advantage of roughly 1.2. However, the effect size was very small for all features.

**2.3.3 Effect of aging.** The absence of a main effect of age for all features is consistent with previous work investigating auditory streaming for older adults with or without hearing loss [19–22]. The absence of a main effect of PTA, despite the older adults having significantly higher audiometric thresholds than the younger adults (see Table 1), suggests that age-related changes in hearing abilities do not negatively impact auditory streaming. These previous studies investigated auditory streaming based on frequency [19–21]; our results extend these findings to intensity and two factors affecting timbre, spectral envelope and temporal envelope. We were especially interested in the interaction between age and musicianship, where a significant interaction would suggest that younger and older adults' performance on this auditory streaming task is differently affected by musicianship. This interaction was not found for any feature, suggesting that younger and older adults' performance is similarly affected by their musical background.

# 3 Experiment 2

To compare between features in their effects on streaming, a dissimilarity paradigm experiment was conducted to establish a common perceptual scale (7; Experiment 2).

## 3.1 Methods

**3.1.1 Participants.** Twelve participants took part in this second experiment, 6 younger and 6 older; 10 also took part in Experiment 1 while the rest were lab members. Table 5

**Table 5. Participant demographics–Experiment 2.**

|  | Age[a] | Gold-MSI training sub-scale score[b] | Pure-tone Average (dB HL)[c] |
|---|---|---|---|
| Younger Adults | 27 (6.4) | 20.2 (10.78) | 2.3 (4.4) |
| Older Adults | 71 (6.0) | 34.8 (3.1) | 15.4 (9.6) |

[a]$t(10) = -12.21$, $p < .01$; standard deviation in brackets

[b]$t(10) = -3.20$, $p > 0.01$; standard deviation in brackets

[c]Better ear average of pure-tone threshold at 500, 1000, 2000 & 4000 Hz; $t(7) = -3.05$, $p = .01$; standard deviation in brackets

outlines this participant pool's demographic information. Level of *musical training* happened to coincide with *age group*, where all *younger adults* except one had <50% Gold-MSI subscale scores, while all *older adults* had >50% scores. This confound is not desirable but occurred because only those willing to stay for Experiment 2 were tested; this happened to produce these homogenous groups. However, as will be demonstrated below, there was no difference in performance between the groups and data were pooled for further analysis.

**3.1.2 Stimuli.** Fifteen four-note melodies were used in this experiment, the same as for Marozeau et al [7], Experiment 2, containing the same F0s as the target melody (G4, C5, A5, D5) in Experiment 1 with all three acoustic features modified simultaneously. As the similarity results were analyzed using multidimensional scaling (MDS), it was important to ensure that the differences in each feature induced perceptual changes that were on the same magnitude scale, and all the stimuli were evenly distributed in a three-dimensional space. For each feature, five possible levels were selected, spanning approximately the upper half of each psychometric function found in Marozeau et al. [7], Experiment 1. Thus, stimuli were presented at loudness levels ranging from 0 to -8 phons of overall attenuation, with 1.69, 1.19, 0.75, 0.35 or 0 dB of attenuation per harmonic and a FHDM (full duration half maximum) value of 100, 112, 126, 142 or 160 ms (see Fig 3). The first five stimuli were constructed by assigning a random permutation of the five levels for each of the three features; this was repeated two more times while ensuring that none of the 15 stimuli were identical. The stimuli were created using

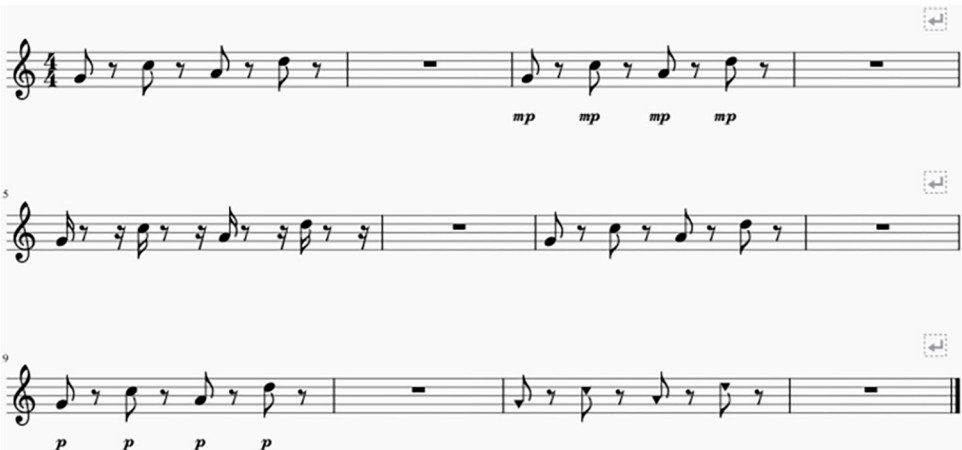

**Fig 3. Illustration of stimuli for Experiment 2.** Example of three possible pairs of stimuli. In the first line, the listener is asked to judge the dissimilarity of the standard melody (bar 1) with the same melody in which each note is reduced in loudness (bar 3). In the second line, the listener is asked to judge the dissimilarity of the melody (bar 5) in which each note is now more impulsive than the standard melody (bar 7). In the third line, the listener is asked to judge the dissimilarity of a soft melody (bar 9) with a melody with a different spectral centroid (bar 11).

Matlab 7.5 and the experiment was implemented in MAX/MSP 8. Stimuli were presented through over-ear headphones (Sennheiser HDA 200) at a comfortable level.

**3.1.3 Procedure.** This experiment was divided into two parts. In the first part, participants could listen to each of the 15 four-note melodies as many times as they wanted in order to acquaint themselves with the range of dissimilarity among the stimuli. In the second part, they rated pairs of melodies on a slider labelled "very dissimilar" at one end and "very similar" at the other, for a total of 105 pairs. Participants rated one pair of melodies at a time and could listen to that pair as many times as they wanted until they were satisfied with their rating (see Fig 3). When then clicked a button to submit their rating, the next trial was triggered. Participants were encouraged to use the full range of the scale. Each response was quantified as an integer ranging from 0 to 128, based on the slider's position.

**3.1.4 Analysis.** MDS analyses were performed in Matlab with custom made functions. Linear modelling was performed in R 3.3.2 using the *lme4* package [65]. Alpha was set at .01, with the conservative Bonferroni correction applied for multiple comparisons, and effect sizes are reported for all statistical tests.

To test for an effect of age, two MDS bootstrap analyses [70] were performed on the data for the older and younger adults groups separately, allowing statistical comparison between the two solutions. In other words, 200 3-dimensional MDS spaces were created by randomly selecting six participants, with replacement (the same participant can be selected many times). If all participants were always in close agreement, the 200 spaces should be very similar. On the other hand, if the participants disagreed, each of the 200 spaces will depend strongly on which participants were randomly selected. Therefore, this technique allows an estimate of the stability of each stimuli according to the inter-participant variability. Two-hundred random selections were used to be consistent with previous research using the same technique [7]. Each space was rotated towards the same reference space created according to the physical properties of the stimuli. These 200 solutions defined a distribution of positions within the MDS space for each stimulus, making it possible to define the 95% confidence volume for each stimulus. To measure similarity between the bootstrap solution for the older and younger adults, a t-test was performed on the position of each stimulus. This test compared the difference of absolute value between the position of a stimulus in the space obtained with the young participants with the position of the same stimulus in the space obtained with the older participants. The standard deviation of the position was extracted based on the bootstrap analysis, and the degrees of freedom based on the number of participants in each group. Bonferroni correction for multiple comparisons was applied. No stimulus reached a p-value close to a significant level. Therefore, the spaces were considered similar and the data from the young and older participants were combined to create a single space that was used for the rest of the analysis.

The similarity ratings were averaged across participants and a three-dimensional space was extracted using the MDSCAL procedure, implemented according to the SMACOFF algorithm [71]. As the MDSCAL solution is rotationally undetermined, the solution was rotated with a procrustean procedure that minimized the least-squares fit between the perceptual and physical spaces.

## 3.2 Results

To determine the amount of dissimilarity between each level of the physical feature, the slope of the regression line between each physical feature and the MDS dimension was plotted (Fig 4). As expected, all three MDS dissimilarity dimensions were correlated with the physical dimensions (Fig 4): the first dimension was correlated with the temporal envelope values, *r*

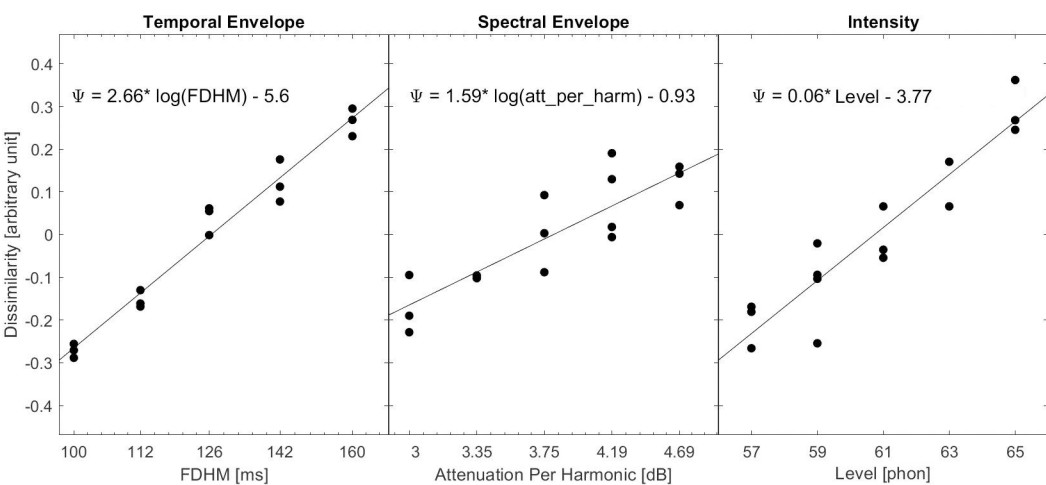

**Fig 4. Scatterplot of the perceptual dimensions derived from the MDS analysis against the physical features.** The equation in each panel describes the regression line. Logarithmic functions are used for temporal and spectral envelope as in Marozeau et al. [7], Fig 8; they offer better correlation to the perceptual dimensions than linear functions.

(11) = 0.93, 95% CIs [0.81, 0.97], *p < 0.01*; the second dimension was correlated with the spectral envelope values, $r$ (11) = 0.86, 95% CIs [0.63, 0.95], *p < 0.01*; and the third dimension was correlated with the intensity values, $r$ (11) = 0.98, 95% CIs [0.95, 0.99], *p < 0.01*.

Based on these relationships, Fig 2 was redrawn with a new scale on the x-axis: Figs 5 and 6 illustrate performance for all three features on the same x-axis dissimilarity scale, for each participant group. The data were re-analyzed in terms of dissimilarity instead of difficulty level. *Dissimilarity* was treated as a continuous variable and all other terms remained the same, including factors and their base categories. Table 6 presents a summary of the model including only predictors with $R^2$ greater than .01 for brevity. Full model specifications can be found in the S1 Appendix. This model included significant main effects of *dissimilarity*, *feature* and *musical training*, but not *age group* or *PTAv*. Both dissimilarity and musical training had positive coefficients (1.33 and 0.07, respectively), indicating better performance with increased dissimilarity and musical training, as expected. There were several significant interactions: *dissimilarity x feature*, *dissimilarity x musical training*, *dissimilarity x feature x musical training*, *feature x age group*, *feature x musical training*, and *feature x age group x musical training*. The interaction between age group and musical training was not significant.

Interactions including dissimilarity indicate differences in slope between features, as indicated by the *dissimilarity* x *feature* two-way interaction, and as a factor of musical training, as indicated by the *dissimilarity* x *feature* x *musical training* three-way interaction. According to model coefficients, slopes for both the spectral and temporal envelope manipulations were shallower than the slope for the intensity manipulation, with spectral envelope having the shallowest slope. This difference in slopes further interacted with musical training, where very small but positive coefficients indicate that higher Gold-MSI scores were associated with slightly steeper slopes.

To confirm the null effect of age, a TOST comparing younger and older adults with lower and upper equivalence bounds of *d'* = -0.42 and *d'* = 0.42, equivalent to a Cohen's *d* of 0.2. The equivalence test was significant, $t$ (6300) = -4.86, p < .01, indicating that the effect, *Δd'* = 0.16 CIs [0.08, 0.25], was different from zero, where younger adults had higher *d'* scores than older adults. However, the effect and its CIs fall within the equivalent bounds, meaning that the effect is also equivalent to zero. In other words, the effect is not bigger than the smallest interesting effect size and therefore can be considered negligible.

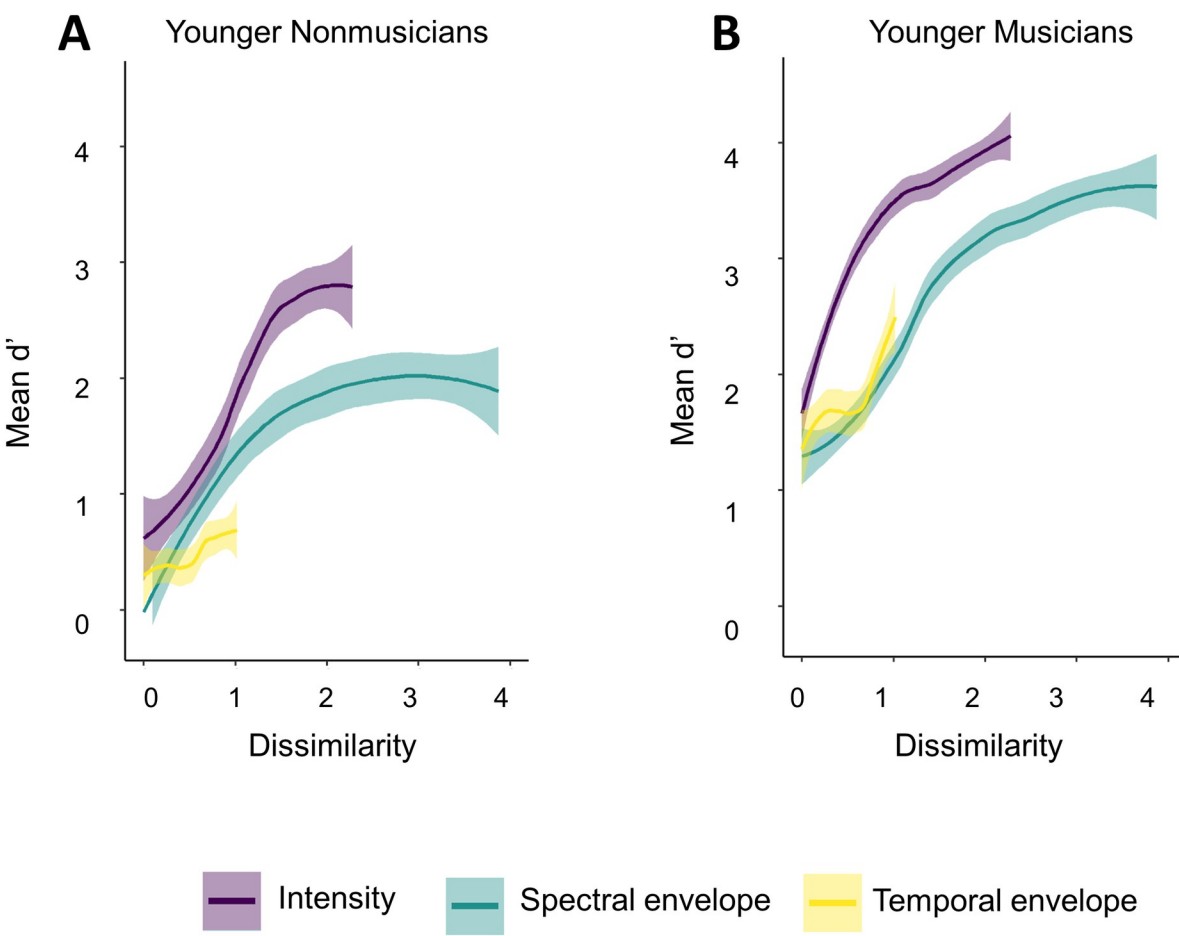

**Fig 5. Mean d' plotted on a common perceptual scale, younger adults.** Performance plotted against dissimilarity, allowing features to be directly compared for younger adults: non-musicians (A) and musicians (B). Standard errors are shown by grey shading around each line.

### 3.3 Discussion

The redrawing of the results of Experiment 1 on a dissimilarity scale in Figs 5 and 6 reveals that each feature spans a different space on the dissimilarity scale, from temporal envelope with the shortest span to spectral envelope with the widest. This suggests that the perceptual distance between 3 dB of harmonic attenuation and 25 dB of harmonic attenuation is four times larger than the perceptual distance between an FHDM of 160 ms and an FHDM of 60 ms, while the perceptual distance between 65 phons and 2 phons is about twice as large. Additionally, the maximally different spectral envelopes used in the study were perceived to be approximately twice as dissimilar as the maximally different intensities. Despite this, *d'* was higher overall when intensity was manipulated, suggesting that loudness is the most effective of these three cues. This is consistent with Marozeau et al.'s [7] results.

**3.3.1 Effect of musicianship.** Marozeau et al. [7] also reported an effect of musicianship, where intensity and spectral envelope required similar dissimilarity to segregate the target from the distractor in non-musicians, while intensity required less dissimilarity than both spectral and temporal envelope to successfully segregate the target from the distractor in musicians. Here, all participants had the least dissimilarity when intensity was manipulated than spectral envelope, although a higher musical training score was associated with needing less dissimilarity to identify deviants for both features. Specifically, musical training had an overall positive

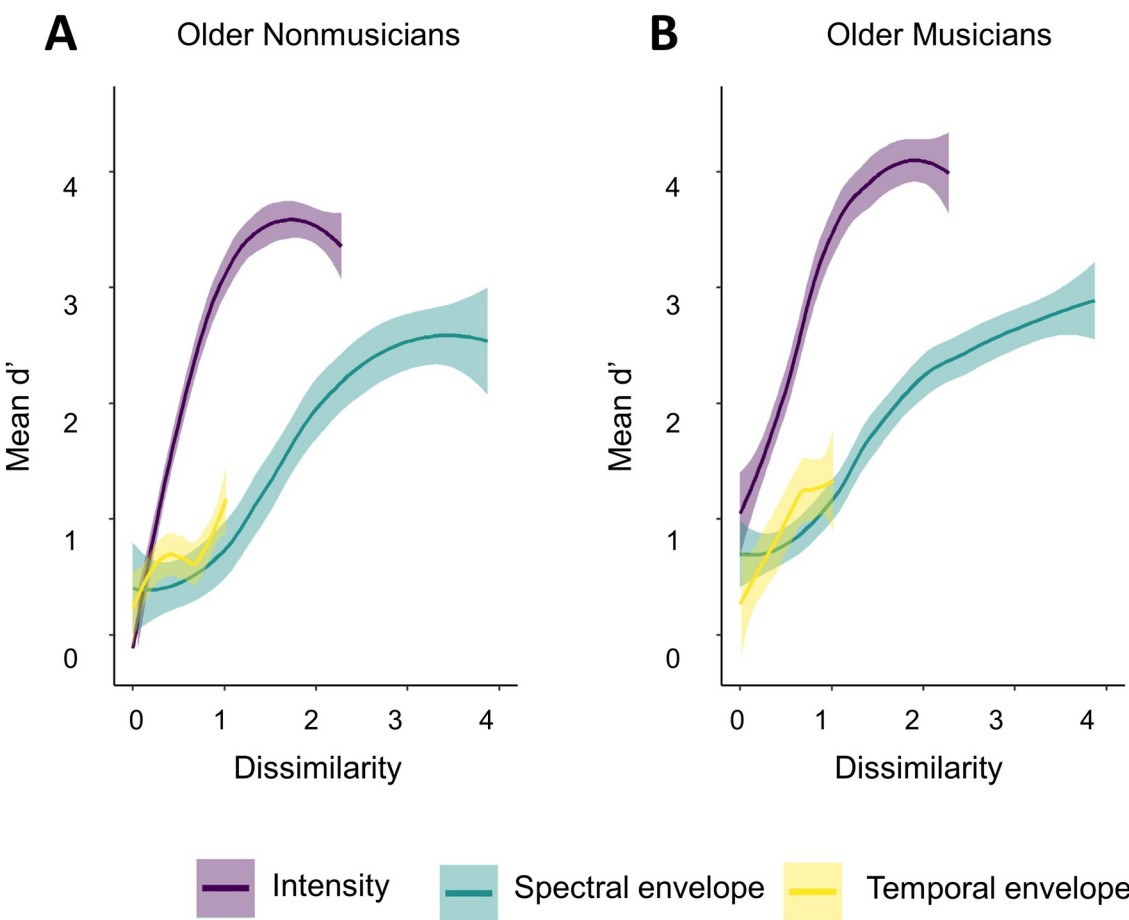

**Fig 6. Mean d' plotted on a common perceptual scale, older adults.** Performance plotted against dissimilarity, allowing features to be directly compared for older adults: non-musicians (A) and musicians (B). Standard errors are shown by grey shading around each line.

coefficient, providing a performance advantage of approximately 1.2 dissimilarity units for musicians with the highest Gold-MSI scores as compared to non-musicians with the lowest Gold-MSI scores. However, note once again that the effect size for musical training was small.

**3.3.2 Effect of aging.** There were no main effects of age or PTA on the relative effectiveness of the three features; however, age interacted with feature, where older adults performed better than younger adults when intensity was manipulated but worse when spectral envelope was manipulated. In other words, older adults relied more heavily on loudness than spectral envelope, while younger adults used both cues. This decreased use of spectral envelope in older adults may be due to older adults' loss of high frequency hearing, which would impede their ability to perceive the manipulations in amplitude envelope that focused on higher frequency attenuation [39, 40] and thus rely more heavily on perceptual loudness cues. A more systematic variation of hearing loss as it interacts with age (i.e. younger and older adults with or without hearing loss) may have helped differentiate between effects of age and hearing loss. However, the results of our current study, as designed, suggest that a different explanation must be sought, as PTA was not a significant predictor of performance. One such explanation may be loudness recruitment, where above a certain intensity, sounds are perceived as louder than for a person with normal hearing [72]. However, we cannot test this explanation with these data as the rate of increase of loudness as a function of increasing level (loudness

**Table 6. Summary of mixed effects multiple linear regression model with common perceptual scale.** Coefficients for each predictor (and each level, as relevant) along with standard error (SE) and predictor $R^2$ are reported along with Pearson's correlation, 95% CIs, p-value and $R^2$ for full models in the note below.

| Predictor | Coefficient | SE | $R^2$ |
|---|---|---|---|
| Intercept | -0.43 | 0.46 | .26 |
| Dissimilarity | 1.33 | 0.14 | .09 |
| Spectral envelope | 0.31 | 0.26 | .13 |
| Temporal envelope | -0.20 | 0.29 | |
| Musical training | 0.06 | 0.02 | < .01 |
| Dissimilarity x Spectral envelope | -0.82 | 0.17 | .01 |
| Dissimilarity x Temporal envelope | -0.74 | 0.36 | |
| Spectral envelope x Age Group | -0.84 | 0.40 | .01 |
| Temporal envelope x Age Group | 0.44 | 0.43 | |
| Dissimilarity x Spectral envelope x Musical training | 0.01 | 0.005 | < .01 |
| Dissimilarity x Temporal envelope x Musical training | 0.02 | 0.01 | |

Note: r = .71, CIs [.70, .72], p < .01, $R^2$ = .51

recruitment) was not collected. Recall that Experiment 1 suggested no difference in performance between younger and older adults when spectral envelope was manipulated; Experiment 2 suggests that though raw performance was comparable, the relative weight of the feature differed between age groups, highlighting the importance of being able to compare the effect of features directly and suggesting that there may be qualitative rather than quantitative age differences in auditory streaming skills. In other words, younger and older adults can perform the task equally well, but the strategies they use to accomplish it differ. Temporal envelope was the feature with the lowest performance for all age groups, suggesting that it is not a particularly useful streaming cue, especially if intensity and spectral envelope information are available.

## 4 General discussion

The experiments reported here provide evidence that sequential auditory streaming, when cued by intensity, spectral envelope and temporal envelope is preserved in older adults who have normal hearing or mild hearing loss, based on the World Health Organization (WHO) criteria [62]. This extends existing evidence for the preservation of sequential auditory streaming for both normal-hearing and hearing-impaired older adults based on frequency and interaural time differences [19–21, 45]. The dissimilarity rating paradigm allowed direct comparison of the effectiveness of the perceptual cues between the groups of participants. Interestingly, intensity was more effective for older adults than for younger adults, although the salience of intensity could be associated with elevated audiometric thresholds, or loudness recruitment, [72–75], and not aging per se for older adults. In the following sections, the effect of musical training is reviewed and results are discussed in the context of theories of aging.

### 4.1 Effect of musicianship

Musicianship is associated with enhanced auditory processing [49, 50, 53, 76–78] including auditory streaming [12]. Enhanced auditory processing abilities are preserved in older musicians compared to older non-musicians [52, 54, 55] while auditory stream segregation based on F0 is preserved in older adults regardless of musicianship [19, 20]. Therefore, one critical question to ask is how musicianship and aging interact during auditory stream segregation

tasks based on intensity, spectral and temporal envelope cues. Significant interactions may suggest a pattern of differential preservation, if aging results in lower *d'* for those with low musical training but not those with high musical training. Main effects of age group and musicianship without an interaction would suggest preserved differentiation, where aging has a negative effect on the *d'* for all participants, regardless of their musical experience [52]. There were no significant interactions between age group and musicianship for any feature, nor when a common perceptual scale was applied. Furthermore, there was no main effect of age group, suggesting that while musical training has a positive effect on *d'*, aging has no negative effect. Combined with the results of Experiment 2, which show that intensity is the most effective perceptual cue for all participant groups, this pattern of results suggests that musicianship is associated with increased sensitivity to acoustic features that are typically less salient for stream segregation. An important caveat is the small effect size of musicianship. This suggests that the positive relationship between musical training and auditory streaming skills is negligible and therefore does not make a significant difference in day to day listening contexts. Indeed, Madsen et al. [79] found that while musical training was associated with better frequency discrimination, interaural time difference discrimination and attentive tracking, this advantage did not extend to speech-in-noise perception. At the same time, Madsen et al. [79] may be an outlier, as a recent meta-analysis revealed that musical training was associated with enhanced abilities to understand speech-in-noise [80]. This pattern of results suggests that musician benefits for understanding speech-in-noise may not be related to enhanced auditory stream segregation. Future meta-analyses of existing work studying the effect of musicianship on auditory streaming would be useful to better understand the importance of the benefit of musicianship.

## 4.2 Theories of aging

One way to consider the streaming task used in the current study is as an inhibition task. Processing of the distractor tones must be inhibited so that the target tones can be integrated into an attended auditory stream. A recent meta-analysis investigated the impact of aging on three types of cognitive inhibition [60]. The meta-analysis found support for age-related decline in the ability to inhibit a natural, habitual or dominant response in favour of a response appropriate to the goal of the study (e.g. Stroop task). For example, in a stop-signal task in which participants inhibited or executed responses based on a visual signal, older adults were less able to inhibit a response to a non-target stimulus [81]. Interestingly, the same meta-analysis found no impact of aging on the ability to inhibit distracting perceptual information, nor the ability to inhibit response interference. For example, Hsieh & Fang [82] found that older adults performed similarly to younger adults on a series of tasks testing participants' ability to ignore distracting information. In this task, participants were asked to report either the direction of a target arrow on the screen (pro condition) or the opposite direction from the target arrow on the screen (anti condition), using the keyboard. The target arrow was surrounded by arrows pointing in the same or the opposite direction in congruent and incongruent conditions, or squares in a neutral condition. In a picture-word Stroop, or response inhibition, task, Bugg [83] found that older adults, like young adults, showed less interference for mostly incongruent items than for mostly congruent items, supporting evidence for intact and flexible reactive control. In the present context, an auditory stream segregation task would be a task that requires the inhibition of distracting information. Accordingly, the findings from the current study are consistent with this meta-analysis, finding no overall deficit in older adults.

There is evidence that although behavioural performance is preserved, brain function may be altered with age. This is known as the compensatory theory of aging [84–86]. Typical

changes include increased frontal lobe activation and decreased hemispheric asymmetry, where instead of more activity in one hemisphere than the other, the two are used equally. In both cases, older adults require the recruitment of more neural resources to accomplish the same task than younger adults. In terms of music perception, such compensatory activity as described above has been observed in the perception of F0 structure [87, 88] and F0-based sequential auditory streaming [20]. Given that differences in the sources of neural activity associated with streaming between younger and older adults have been observed for F0-based auditory stream segregation [20], it is possible that similar mechanisms were at play in the current study. Future research should examine potential compensatory activity using brain imaging methods during sequential auditory streaming based on acoustic cues other than F0.

In conclusion, this study investigated auditory stream segregation based on intensity, spectral and temporal envelope cues for older and younger musicians and non-musicians. The findings confirm that sequential auditory streaming is generally preserved in older adults, and enhanced in musicians. Furthermore, older adults relied on intensity more than younger adults and musicians were better able to use a variety of acoustic cues than non-musicians.

## Supporting information

**S1 Appendix. Full model and t-test details for all linear modelling.**
(DOCX)

**S2 Appendix. Summary statistics (mean, standard deviation) of d' and transformed d' by level, age group, musicianship, and feature.**
(DOCX)

## Acknowledgments

The authors acknowledge that this work was undertaken on Ktaqmkuk, the unceded, unsurrendered lands of the Mi'kmaq and the ancestral lands of the Beothuk. We would also like to acknowledge the Inuit of Nunatsiavut and NunatuKavut and the Innu of Nitassinan, and their ancestors, as the original peoples of Labrador. The authors thank Alex Cho and Liam Foley for their help with participant recruitment and data collection.

## Author Contributions

**Conceptualization:** Sarah A. Sauvé.

**Data curation:** Sarah A. Sauvé.

**Formal analysis:** Sarah A. Sauvé, Jeremy Marozeau.

**Funding acquisition:** Benjamin Rich Zendel.

**Investigation:** Sarah A. Sauvé.

**Methodology:** Sarah A. Sauvé.

**Project administration:** Sarah A. Sauvé.

**Resources:** Benjamin Rich Zendel.

**Software:** Jeremy Marozeau.

**Supervision:** Sarah A. Sauvé, Benjamin Rich Zendel.

**Visualization:** Sarah A. Sauvé, Jeremy Marozeau.

**Writing – original draft:** Sarah A. Sauvé.

**Writing – review & editing:** Sarah A. Sauvé, Jeremy Marozeau, Benjamin Rich Zendel.

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
