## [Decision Letter · Decision Letter 0]

19 May 2022

PONE-D-22-00276The effects of aging and musicianship on the use of auditory streaming cuesPLOS ONE

Dear Dr. Sauvé,

Thank you for submitting your manuscript to PLOS ONE. After careful consideration, we feel that it has merit but does not fully meet PLOS ONE’s publication criteria as it currently stands. Therefore, we invite you to submit a revised version of the manuscript that addresses the points raised during the review process. Please submit your revised manuscript by Jul 03 2022 11:59PM. If you will need more time than this to complete your revisions, please reply to this message or contact the journal office at plosone@plos.org. Please include the following items when submitting your revised manuscript:A rebuttal letter that responds to each point raised by the academic editor and reviewer(s). You should upload this letter as a separate file labeled 'Response to Reviewers'.A marked-up copy of your manuscript that highlights changes made to the original version. You should upload this as a separate file labeled 'Revised Manuscript with Track Changes'.An unmarked version of your revised paper without tracked changes. You should upload this as a separate file labeled 'Manuscript'.

We look forward to receiving your revised manuscript.

Kind regards,

Qian-Jie Fu, Ph.D.

Academic Editor

PLOS ONE

Journal Requirements:

Reviewers' comments:

Reviewer's Responses to Questions

**Comments to the Author**

1. Is the manuscript technically sound, and do the data support the conclusions?

Reviewer #1: Yes

Reviewer #2: Yes

2. Has the statistical analysis been performed appropriately and rigorously? 

Reviewer #1: I Don't Know

Reviewer #2: Yes

3. Have the authors made all data underlying the findings in their manuscript fully available?

Reviewer #1: Yes

Reviewer #2: No

4. Is the manuscript presented in an intelligible fashion and written in standard English?

Reviewer #1: Yes

Reviewer #2: Yes

5. Review Comments to the Author

Reviewer #1: In their study, the authors examine stream segregation ability in young and older musicians and non-musicians to separate known influences of musicality and age. They examine the role of specific stimulus parameters, such as intensity, and spectral and temporal envelope. They replicate the influence of musicality, but find no general age-related deficit in stream segregation. However, older people are more likely to use the intensity dimension than younger people and musicians, who are more likely to use the entire set of all manipulated dimensions.

The study is interesting both in terms of music perception and (even more generally) stream segregation, a skill that plays an important role in everyday life, and in terms of age and age-related changes in listening and cognitive processing of auditory stimuli. The manuscript is well structured and gives a good theoretical introduction to the topic, the hypotheses are clearly founded and derived from theory. The two experiments are well designed and conducted, the analysis seems appropriate and up-to-date, the results are quite clear and adequately discussed. However, a few points are unclear and should be revised before a decision can be made about publication in PLOSone.

General: The role of hearing loss is unclear. In the introduction, this is mentioned as a possible influencing factor, but then not systematically varied. One possibility could be to treat age and hearing loss as two independent variables, for example, to differentiate younger and older subjects into groups with normal and reduced hearing.

Further comments:

Line 178: The reference to the Gold MSI could be supplemented with a reference to the following chapter 2.1.2, where it is described in more detail.

Table 2: This table is very confusing, perhaps 2 separate tables with one for intensity and a second for spectral envelope would be more convenient.

Line 449: A visualization of the stimuli used in experiment 2 (similar to figure 1) and the experimental procedure would be helpful.

Line 563: "The redrawing of the results of Experiment 1 on a dissimilarity scale in Fig 4 reveals ...". Shouldn't reference be made here to Figs. 4 and 5?

Line 594: "as neither PTA nor PTA8kHz (analysis not reported)" Citing an analysis not reported as evidence seems inappropriate to me. Either the analysis should be described as an appendix or the reference should be deleted.

Figures 2, 4, 5: The difference between the groups is hardly visible, perhaps colors could be used to better differentiate the curves.

Reviewer #2: Thank you for the opportunity to review this manuscript. I just have a few questions / comments.

Line 80 – Low frequency stimuli were used, and this was the region of normal hearing in both young and old groups. The differences in PTA was in the high frequencies.

Line 107 – perhaps higher harmonics are not resolved due to broader auditory filters

Line 256 – I understand that the variable of interest is the intensity difference, but how does one know if the absolute intensity does not affect the baseline performance? For example, I’m thinking of many psychoacoustic studies showing psychometric function, where differing changes along one scale has small or large affects on performance.

Line 340, 398 - This suggested that even at higher difficulties there were differences in performance between musicians and non-musicians, but the model as described in 336 does not include significant interactions. Am I misunderstanding why some models include significant interactions, and others don’t?

Line 423 – Also see Tejani et al, 2017, JASA who didn’t see effects of aging on sequential streaming.

Line 639 - Madsen et al, 2019, Sci Reports is relevant here in that they found musician advantage in psychoacoustic tasks, but it didn’t translate into advantages for speech understanding in noise.

https://www.nature.com/articles/s41598-019-46728-1

6. PLOS authors have the option to publish the peer review history of their article (what does this mean?). If published, this will include your full peer review and any attached files.

Reviewer #1: No

Reviewer #2: No

---

## [Author Response · Author response to Decision Letter 0]

27 Jul 2022

Reviewer #1: In their study, the authors examine stream segregation ability in young and older musicians and non-musicians to separate known influences of musicality and age. They examine the role of specific stimulus parameters, such as intensity, and spectral and temporal envelope. They replicate the influence of musicality, but find no general age-related deficit in stream segregation. However, older people are more likely to use the intensity dimension than younger people and musicians, who are more likely to use the entire set of all manipulated dimensions.

The study is interesting both in terms of music perception and (even more generally) stream segregation, a skill that plays an important role in everyday life, and in terms of age and age-related changes in listening and cognitive processing of auditory stimuli. The manuscript is well structured and gives a good theoretical introduction to the topic, the hypotheses are clearly founded and derived from theory. The two experiments are well designed and conducted, the analysis seems appropriate and up-to-date, the results are quite clear and adequately discussed. However, a few points are unclear and should be revised before a decision can be made about publication in PLOSone.

General: The role of hearing loss is unclear. In the introduction, this is mentioned as a possible influencing factor, but then not systematically varied. One possibility could be to treat age and hearing loss as two independent variables, for example, to differentiate younger and older subjects into groups with normal and reduced hearing.

We do not have the data to properly address the hearing loss issue, because none of our participants met the criteria for hearing loss (>25 dB HL PTA). Accordingly, in this data set Aging and PTA effectively measure the same thing. Future work should recruit participants with hearing loss in both younger and older adults to properly tease this apart. We have added a sentence in the introduction and discussion acknowledging that age and hearing loss are independently important, but that hearing loss was not systematically varied.

Line 162-163: “Accordingly, both age and hearing loss will be important variables to consider independently.”

Line 608-612: “A more systematic variation of hearing loss as it interacts with age (i.e. younger and older adults with or without hearing loss) may have helped differentiate between effects of age and hearing loss. However, the results of our current study, as designed, suggest that a different explanation must be sought […]”

Further comments:

Line 178: The reference to the Gold MSI could be supplemented with a reference to the following chapter 2.1.2, where it is described in more detail.

Reference to section 2.1.2 has been added.

Table 2: This table is very confusing, perhaps 2 separate tables with one for intensity and a second for spectral envelope would be more convenient.

Table 2 is now Tables 2 and 3.

Line 449: A visualization of the stimuli used in experiment 2 (similar to figure 1) and the experimental procedure would be helpful.

We have created a figure (Fig 3) to describe the stimuli used Experiment 2 as well as examples of pair judgements.

Line 563: "The redrawing of the results of Experiment 1 on a dissimilarity scale in Fig 4 reveals ...". Shouldn't reference be made here to Figs. 4 and 5?

Yes, thank you, reference is now to both figures. (Line 579)

Line 594: "as neither PTA nor PTA8kHz (analysis not reported)" Citing an analysis not reported as evidence seems inappropriate to me. Either the analysis should be described as an appendix or the reference should be deleted.

Reference to the unreported analysis has been removed.

Figures 2, 4, 5: The difference between the groups is hardly visible, perhaps colors could be used to better differentiate the curves.

Figures have been redone with a colour-blind friendly palette.

Reviewer #2: Thank you for the opportunity to review this manuscript. I just have a few questions / comments.

Line 80 – Low frequency stimuli were used, and this was the region of normal hearing in both young and old groups. The differences in PTA was in the high frequencies.

We have modified this sentence so that it now reads:

Line 79-82: “Despite these PTA differences between older and younger adults, there was no difference in streaming perception between the groups. This may be because the stimuli were all below 1000 Hz, and both groups of participants had normal thresholds at those frequencies. ”

Line 107 – perhaps higher harmonics are not resolved due to broader auditory filters

This possibility has been added:

Line 108-110: “It is possible that these higher harmonics were not resolved due to broader auditory filters (40), which are specifically associated with hearing loss rather than increased age (41).”

Line 256 – I understand that the variable of interest is the intensity difference, but how does one know if the absolute intensity does not affect the baseline performance? For example, I’m thinking of many psychoacoustic studies showing psychometric function, where differing changes along one scale has small or large affects on performance.

This is a good question. With our design, it is not possible to know how absolute intensity may have affected performance. As far as we know, no study has compared stream segregation at different overall intensities; intensity is always manipulated. We tested for differences in performance according to the dB at which the stimuli were set to present and found no effect. It is a caveat now explicitly acknowledged in the Stimuli section and the discussion:

Line 258-259: “This affects the absolute intensity at which the stimuli were presented, which may affect baseline performance […]”

Line 413-416: “Performance was also not significantly related to differences in the intensity at which stimuli were presented through the headphones, suggesting that there are not significant differences in baseline performance as a function of absolute intensity.”

Line 340, 398 - This suggested that even at higher difficulties there were differences in performance between musicians and non-musicians, but the model as described in 336 does not include significant interactions. Am I misunderstanding why some models include significant interactions, and others don’t?

The text at the locations identified does not describe an interaction but a main effect, highlighting that the effect is maintained even as the task gets more difficult. To clarify, the sentences have been simplified and now read as follows:

Line 349-350: “Participants with higher musical training scores were better at detecting deviants overall.”

Line 405-407: “More importantly, people with higher musical training scores were better able to make use of all features of the distractor at all levels of difference from the target.”

Line 423 – Also see Tejani et al, 2017, JASA who didn’t see effects of aging on sequential streaming.

This reference has been added to the introduction and discussion.

Line 639 - Madsen et al, 2019, Sci Reports is relevant here in that they found musician advantage in psychoacoustic tasks, but it didn’t translate into advantages for speech understanding in noise. https://www.nature.com/articles/s41598-019-46728-1

A sentence summarizing this finding has been added:

Line 658-664: “Indeed, Madsen et al. (80) found that while musical training was associated with better frequency discrimination, interaural time difference discrimination and attentive tracking, this advantage did not extend to speech-in-noise perception. At the same time, Madsen et al. (80) may be an outlier, as a recent meta-analysis revealed that musical training was associated with enhanced abilities to understand speech-in-noise (81). This pattern of results suggests that musician benefits for understanding speech-in-noise may not be related to enhanced auditory stream segregation.”

---

## [Decision Letter · Decision Letter 1]

1 Sep 2022

The effects of aging and musicianship on the use of auditory streaming cues

PONE-D-22-00276R1

Dear Dr. Sauvé,

We’re pleased to inform you that your manuscript has been judged scientifically suitable for publication and will be formally accepted for publication once it meets all outstanding technical requirements.

Kind regards,

Qian-Jie Fu, Ph.D.

Academic Editor

PLOS ONE

Additional Editor Comments (optional):

Reviewers' comments:

Reviewer's Responses to Questions

**Comments to the Author**

1. If the authors have adequately addressed your comments raised in a previous round of review and you feel that this manuscript is now acceptable for publication, you may indicate that here to bypass the “Comments to the Author” section, enter your conflict of interest statement in the “Confidential to Editor” section, and submit your "Accept" recommendation.

Reviewer #1: All comments have been addressed

Reviewer #2: All comments have been addressed

2. Is the manuscript technically sound, and do the data support the conclusions?

Reviewer #1: Yes

Reviewer #2: Yes

3. Has the statistical analysis been performed appropriately and rigorously? 

Reviewer #1: Yes

Reviewer #2: Yes

4. Have the authors made all data underlying the findings in their manuscript fully available?

Reviewer #1: Yes

Reviewer #2: No

5. Is the manuscript presented in an intelligible fashion and written in standard English?

Reviewer #1: Yes

Reviewer #2: Yes

6. Review Comments to the Author

Reviewer #1: The authors have taken all my comments into account and I recommend that the manuscript be accepted for publication in PLOSone.

One single small typo that I noticed:

Line 492: “When then clicked a button to submit their rating, … “ should probably be “When they clicked a button to submit their rating, …”

Reviewer #2: The authors have satisfactorily addressed my concerns. Thank you for the opportunity to review this manuscript.

7. PLOS authors have the option to publish the peer review history of their article (what does this mean?). If published, this will include your full peer review and any attached files.

Reviewer #1: No

Reviewer #2: No

---

## [Editor Report · Acceptance letter]

12 Sep 2022

PONE-D-22-00276R1 

The effects of aging and musicianship on the use of auditory streaming cues 

Dear Dr. Sauvé:

I'm pleased to inform you that your manuscript has been deemed suitable for publication in PLOS ONE. Congratulations! Your manuscript is now with our production department. 

Kind regards, 

on behalf of

Dr. Qian-Jie Fu 

Academic Editor

PLOS ONE